# Central Serous Chorioretinopathy by Autofluorescence, Enface and SLO–Retromode Imaging

**DOI:** 10.3390/life13061407

**Published:** 2023-06-17

**Authors:** Maria Cristina Savastano, Claudia Fossataro, Riccardo Sadun, Andrea Scupola, Maria Grazia Sammarco, Clara Rizzo, Pia Clara Pafundi, Stanislao Rizzo

**Affiliations:** 1Ophthalmology Unit, Fondazione Policlinico Universitario A. Gemelli, IRCCS, 00168 Rome, Italy; mariacristina.savastano@gmail.com (M.C.S.); fossataroclaudia@gmail.com (C.F.); stanislao.rizzo@unicatt.it (S.R.); 2Ophthalmology Unit, Catholic University of the Sacred Heart, 00168 Rome, Italy; 3Ocular Oncology Unit, Fondazione Policlinico Universitario A. Gemelli, IRCCS, 00168 Rome, Italy; scupola99@gmail.com (A.S.); mariagrazia.sammarco@policlinicogemelli.it (M.G.S.); 4Ophthalmic Unit, Department of Neurosciences, Biomedicine and Movement Sciences, University of Verona, 37129 Verona, Italy; clararizzo2@gmail.com; 5Facility of Epidemiology and Biostatistics, Gemelli Generator, Fondazione Policlinico Universitario A. Gemelli, IRCCS, 00168 Rome, Italy; piaclara.pafundi88@gmail.com; 6Istituto di Neuroscienze del CNR, 56124 Pisa, Italy

**Keywords:** autofluorescence, central serous chorioretinopathy, enface, retromode imaging

## Abstract

The aim of our study was to investigate the clinical features of central serous chorioretinopathy (CSC) with autofluorescence (AF), retromode (RM), and enface imaging. This retrospective study was conducted at Fondazione Policlinico Universitario A. Gemelli, IRCCS, Rome (Italy), between September and December 2022. Each patient underwent a complete ophthalmological examination, which included optical coherence tomography (OCT), enface image analysis, AF, and RM imaging. We further evaluated the presence and area of extension of serous retinal detachment and retinal pigment epithelium (RPE) atrophy through AF, RM, and enface imaging. We included 32 eyes from 27 patients (mean age: 52.7 ± 13.3 years). The median AF area was 19.5 mm^2^ (IQR 6.1–29.3), while the median RM area was 12.3 mm^2^ (IQR 8.1–30.8), and the median enface area was 9.3 mm^2^ (IQR 4.8–18.6). RPE atrophy was diagnosed in 26 cases (81.3%) with RM imaging and in 75% of cases with AF. No difference emerged between AF and RM analysis in the detection of central serous detachment in CSC. However, RM imaging showed a high specificity (91.7%) and negative predictive value (84.6%) to detect RPE changes when compared to the AF standard-of-care technique. Thus, RM imaging could be considered an adjunctive imaging method in CSC.

## 1. Introduction

Central serous chorioretinopathy (CSC) is a common retinal disease that preferentially affects young or middle-aged men and is characterized by a localized serous retinal detachment, retinal pigment epithelium (RPE) atrophy and, sometimes, pigment epithelium detachment (PED) [1,2]. The most common area of occurrence is the macula; however, serous detachment, usually asymptomatic, can also occur in other extramacular regions [2]. Described for the first time by von Graefe in 1866, there is still debate on CSC’s pathogenesis [3]. This entity has been associated with corticosteroid therapy; type A personality; sleep alteration, particularly with obstructive sleep apnea syndrome (OSAS); and gastrointestinal disturbance [2]. 

CSC belongs to a new group of disorders that are defined as pachychoroid diseases (PCDs). PCDs were first proposed by Warrow et al. in 2013 and characterized by morphological changes and the dysfunction of the choroid [4]. This group includes several chorioretinal disorders, although CSC and polypoidal choroidal vasculopathy (PCV) are the two predominant PCD phenotypes. The pathogenesis of CSC, in addition to other PCDs, remains a significant challenge and, thus, deserves further investigation. Several studies have shown a correlation between CSC in pathologically dilated Sattler and Haller vessels with zones that have a reduced choriocapillaris (CC) flow and the resulting activation of collateral circulation [5]. Moreover, a rearrangement of the morphological pattern in the Haller layer in CSC eyes compared to healthy eyes suggested a hemodynamic implication in deeper choroidal vasculature [6,7]. 

After its first appearance, CSC is often followed by a complete and spontaneous resolution within a few weeks or months, and it is considered acute if it lasts less than six months. Patients are typically referred during the acute phase for central visual acuity impairment, relatively central scotoma, metamorphopsia, dyschromatopsia, and reduced contrast sensitivity [8]. 

CSC is considered chronic if clinical manifestations do not resolve within six months. The chronic form is defined as diffuse retinal epitheliopathy, in which alterations of the RPE profile and its rearrangement are observed over time. These alterations are associated with either focal or diffuse areas of RPE atrophy, which appear on structural OCT as the absence of RPE with a backscattering effect [9]. Photodynamic therapy (PDT) has been used to treat chronic CSC for several years, consisting of laser activation in the photosensitizer verteporfin and focusing on the smokestack found by fluorescein angiography (FA), resulting in a decrease in choroidal hyperpermeability [10]. In view of the high corticosteroid and mineralocorticoid levels in CSC patients, eplerenone, a mineralocorticoid antagonist administered orally once per day, has been proposed as a valid therapeutic option [11]. A subthreshold micropulse laser with a 5% duty cycle represents a third therapeutic approach, consisting of an RPE stimulation by multiple laser pulses, favoring subretinal fluid absorption [11].

To correctly diagnose CSC, a multimodal imaging approach is essential, including fundus retinography, autofluorescence (AF), SD-OCT, fluorescein angiography, and indocyanine green (ICG) angiography [12]. Autofluorescence is considered one of the most sensitive methods for identifying CSC. In the acute phase, focal regions of hypoautofluorescence can be observed; these regions could disappear consequently to the resolution of the event, or they could be replaced by hyperautofluorescent areas in unsolved cases [13,14]. Furthermore, considering its noninvasiveness, AF is usually performed during follow-up to analyze changes in the appearance of the disease resulting from different therapeutic strategies, such as pharmacology, laser treatment, and photodynamic therapy [11]. Additionally, the sensitivity of AF is based on its capability to highlight previous RPE involvement in cases of gravitational disease [11,15]. 

Currently, structural OCT has become the cornerstone for both the diagnosis and management of CSC. Since the beginning, pathological features have been only partially appreciable with time-domain devices (TD–OCT). The introduction of spectral domain OCT (SD–OCT) has provided the ability to precisely detect all the characteristic details; however, with the most recent swept-source devices (SS–OCT), we are currently able to define even chorioretinal changes that are smaller than 5 microns. In addition, swept-source OCT shows in fine detail the vortex vein anastomoses at a watershed in pachychoroid spectrum disease, which is sometimes associated with pachyvessels, compared to healthy eyes [15]. The meaning of anastomoses remains unclear. However, compensation for choroidal congestion might establish a new drainage route between the superior and inferior vortex veins.

Retromode imaging (RM), recently introduced, was performed by a scanning laser ophthalmoscope (SLO), which used infrared wavelengths of light to penetrate the deeper retinal layers. Once the retina was highlighted, the central direct reflex was stopped, and the backscattering light was collected by an opening on either the right or left side. The resultant retinal image was characterized by a pseudo-3-dimensional (3D) facet and, consequently, high-contrast margins of retinal alterations, which could lead to a better evaluation of RPE changes, as well as serous retinal detachment in CSC eyes [16,17,18,19,20].

Enface OCT is a noninvasive diagnostic technique that is capable of providing coronal images of the posterior segment at different depth levels, supplying additional information to cross-sectional imaging for a more accurate diagnosis [21].

Thus, the aim of our study was to evaluate CSC eyes with a well-known technique, AF, and this novel imaging method, RM. In addition, when available, we compared AF and RM images with enface scans.

## 2. Materials and Methods

### 2.1. Study Design and Population

This single-center, observational, retrospective study was conducted at Fondazione Policlinico Universitario A. Gemelli, IRCCS, Rome (Italy) between September and December 2022. We enrolled patients with CSC at different stages of the disease, regardless of the performed therapy. Included patients could either be experiencing their first symptomatic manifestation (acute phase) or already be in a chronic stage.

We excluded all patients with optic media opacities, diabetic retinopathy, other vascular retinal diseases, uveitis, myopia higher than 6 diopters, and low-quality images (quality index lower than 7/10). After a full explanation, each patient signed an informed consent form.

The study was conducted in accordance with the 1976 Declaration of Helsinki and its later amendments and was approved by the local ethics committee of Fondazione Policlinico Universitario Agostino Gemelli of Rome (Italy) (ID: 3680).

### 2.2. Variables and Procedures

Each patient underwent a complete ophthalmological examination, including best-corrected visual acuity (BCVA), slit lamp examination, dilated fundus evaluation, structural OCT, OCT angiography (OCTA), enface scans (Solix full-range OCT, Optovue Inc., Freemont, CA, USA), fundus retinography (Nidek Mirante, Nidek Co., Ltd., Gamagori, Japan), autofluorescence (blue-light autofluorescence) (Nidek Mirante, Nidek Co., Ltd., Gamagori, Japan), and retromode imaging (Nidek Mirante, Nidek Co., Ltd., Gamagori, Japan) (Figure 1). Each eye was dilated with 1% tropicamide drops before the exams. High-definition OCT B scans (B scan density 512 × 512, 5-micron resolution) were performed centering on the macula region, passing horizontally and vertically through the fovea. The OCTA protocol (Angio-Vue) consisted of a 6.4 × 6.4 mm scan focused on the fovea. This software was provided with motion correction technology, which corrected segmentation errors due to either eye movements or blinks. Once the exam was performed, the OCTA scan, enface image, and corresponding OCT slab were automatically provided by the software in the same printout. We collected enface images at the outer retina level (see the correspondence between enface and OCT scan in Figure 2).

We further evaluated the extent (area expressed in mm^2^) of the disease features in AF, RM, and enface scans using ImageJ 1.8.0_345 software (ImageJ 1.53 k, National Institutes of Health, Bethesda, MD, USA) (Figure 3 and Figure 4). First, each image was converted to 8 bits, and the scale was set according to the size of the scan in millimeters, which was then converted into pixels. Then, we manually outlined the region of interest (ROI) with the provided caliper tool, and the software provided the area of the ROI. The ROIs were manually selected by two co-authors. A retinal specialist checked the segmented regions once the images were collected. If all three operators agreed, the image was consequently included in the analysis. Cohen’s coefficient analysis was used to ensure a concordance of at least 0.90. A neuroepithelium serous detachment was identified on the enface images as a hyporeflective area; within the enface images, it usually appeared isoautofluorescent. The pseudo-3D effect of retromode imaging helped to outline the detached area. RPE atrophy appeared as hyper- or hypoautofluorescence on AF with hyperreflective areas on enface imaging and mottling on RM imaging.

Moreover, we recorded the presence of subretinal fluid (SRF) and pigment epithelium detachment from the OCT scans. Additionally, from the observation of AF and RM images, we evaluated the presence of RPE atrophy in each patient’s eye. RPE atrophy was evaluated as hyperfluorescence, corresponding to intracellular fluorophore accumulation or hypofluorescence, and caused by nonfunctional RPE cells. Two operators independently analyzed the images, and a retinal expert ophthalmologist validated their data.

### 2.3. Statistical Analysis

The clinical and demographic characteristics of the sample were described by descriptive statistics techniques. In-depth, qualitative data were expressed as absolute and relative percentage frequencies, while quantitative data were expressed as either the mean and standard deviation (SD) or median and interquartile range (IQR), as appropriate. To verify the Gaussian distribution of quantitative variables, the Shapiro-Wilk test was applied. Missing values, <5% in all cases, were treated by appropriate imputation methods implemented with *imputeR* R software, as fully described in the Appendix A.

Comparisons between the different areas were performed using a Wilcoxon rank-sum test for paired data, while the three diagnostic technique outcomes were compared using the McNemar test.

Spearman’s correlation test was applied to assess the correlation between the variables of interest. Spearman’s rho coefficient, 95% confidence interval (CI), and *p*-value were further reported. A heatmap further depicted the observed correlations.

The accuracy of retromode imaging compared to autofluorescence was assessed by ROC curve analysis. The area under the ROC curve (AUROC) and 95% confidence interval, specificity, sensitivity, and negative and positive predictive values (NPV and PPV) were further reported, along with the ROC curve itself.

Statistical significance was set at *p* < 0.05. Suggestive *p* values were also reported (0.05 ≤ *p* < 0.10). All analyses were conducted with R software v4.3.0 (CRAN^®^, R Core 2023, Wien, Austria) [22]. A detailed plan with the R packages used for the abovementioned analyses is reported in the Appendix A.

## 3. Results

We finally included 32 eyes, mostly left eyes (65.6%), from 27 patients, mainly males (77.8% vs. 22.2% females). The mean age was 52.7 ± 13.3 years. The median visual acuity was 0.69 LogMar (0.50–0.90). AF, RM imaging, and structural SD-OCT were performed on all the eyes (n = 32), whereas enface images were available only for 18 eyes due to a lack of high-quality images for the remaining eyes (i.e., at least >7/10, as asserted in the inclusion criteria). This lack of quality may have been due to the longer time needed to perform the enface exam, which could have affected the quality of the scans.

For the diagnostic techniques, the median autofluorescence (AF) area was 19.5 mm^2^ (IQR 6.1–29.3), which was slightly larger than the retromode (RM) imaging area [median 12.3 mm^2^ (IQR 8.1–30.8)], but this difference was not significant (*p* = 0.761). The enface images exhibited a median area of 9.3 mm^2^ (IQR 4.8–18.6). AF and enface significantly differed from one another, with the latter remarkably smaller (*p* < 0.001).

RPE atrophy was diagnosed in 26 cases (81.3%) with RM imaging and in 24 cases (75%) with AF; no significant difference was observed between the two techniques (*p* = 0.683). A further accuracy analysis between these two techniques revealed a moderate accuracy of retromode imaging compared to the gold-standard AF, with an area under the ROC curve of 0.708 (95% CI 0.515–0.902) (Figure 5). AF was used as the reference (diagonal line), i.e., RPE atrophy diagnosis by AF was used to assess the accuracy of diagnosis with the other method, i.e., RM imaging. The specificity of RM imaging was extremely high (91.7%), indicating a high ability to discriminate true-negative cases, with a false-positive rate of 9.3%. However, sensitivity (ordinate axis), which indicates the true-positive rate, was quite low, reaching only a 50% ability to recognize true positives. The positive predictive value (PPV) was 66.7%, and the negative predictive value (NPV) was 84.6%.

Finally, subretinal fluid was detected in 93.7% of the eyes (n = 30), while DEP was detected in 18.8% (n = 6).

All data are reported in Table 1.

We further performed a correlation analysis to better characterize the relationships between the diverse techniques. In particular, a remarkably higher direct correlation between AF and RM areas was confirmed, with a significantly higher correlation coefficient (rho 0.91, 95% CI 0.81–0.96; *p* < 0.001). A similar finding was also observed between AF and enface (rho 0.82, 95% CI 0.64–0.91; *p* < 0.001) and between RM and enface (rho 0.82, 95% CI 0.61–0.94; *p* < 0.001). Of note, enface exhibited a significant inverse correlation with visual acuity, although with a moderate correlation coefficient (rho −0.43, 95% CI −0.68–−0.11; *p* = 0.014), while no significant correlation emerged between visual acuity and either AF or RM.

Among the other correlations analyzed, we found a significant direct correlation for age with all the diagnostic techniques, with correlation coefficients of approximately 0.40 in almost all cases, and with the highest coefficients observed with AF (rho 0.46, 95% CI 0.12–0.72; *p* = 0.007) and enface (rho 0.45, 95% CI 0.08–0.73; *p* = 0.009). An inverse correlation was observed between age and visual acuity (rho −0.63, 95% CI −0.83–−0.36; *p* < 0.001). All data are shown in Appendix A and in the heatmap in Figure 6.

## 4. Discussion

CSC is a pachychoroid disorder and a complex and only partially understood entity that is characterized by the serous detachment of the retina and is associated with the presence of single or multiple areas of choroidal leakage through defects of the RPE [23,24]. However, the pathophysiology and exact sequences of events are still debated. Many authors believe that the disease originates in a choriocapillaris/choroidal dysfunction, which has been well described by indocyanine green angiography (ICGA) and OCTA [25,26]. According to this hypothesis, a delayed choroidal filling may represent the first step, leading to increased pressure and long-standing venous congestion with the dilatation of Haller’s layers. Choriocapillaris ischemia with capillary leakage and dropout would be the natural consequence of these events, followed by RPE dysfunction, serous neurosensory retinal detachment, and atrophic RPE changes [27].

An alternative hypothesis implicates the RPE first, leading to subretinal and intraretinal fluid accumulation and a consequent chronic insult to the RPE itself [28]. However, there is substantial evidence that defects in the RPE are indeed secondary to choroidal dysfunction, as choroidal abnormalities are at least as extensive as RPE abnormalities [25,26]. 

Recently, Mohabati et al. suggested an impairment in the natural pumping function of RPE and Müller cells [29]. They hypothesized that an inability of the RPE to drain the fluid percolating through the retina into the choroid could underlie the pathophysiological mechanism of CSC. Thus, the identification of RPE abnormalities is extremely useful for the diagnosis, choice of treatment, and follow-up of patients with CSC. However, traditional angiographic techniques, such as FA and ICGA, can provide unclear information regarding the outer retina and the RPE. Additional details may be detected by SD-OCT, which is able to detect morphologic alterations in the RPE in most cases, especially those that are dependent on the chronicity of the disease [30,31]. 

In addition, the introduction of OCTA allowed pachychoroid neovasculopathy to be highlighted with the characteristic of tangled filamentous vessels overlying a focal area of the thickened choroid in the CSC [32].

The spectrum of pachychoroid also remains a challenge in its definition, as reported by Spaide, as it includes many different clinical conditions that are similar to those stated elsewhere [33].

Consistent with these challenges in interpreting the state of morphological changes, new noninvasive methods of analysis could help in both the diagnosis and management of the complexity of CSC.

Fundus autofluorescence (AF) is an additional noninvasive technique that provides topographic information regarding RPE metabolism, even though it has limited ability in the anatomic resolution of RPE details [20,34].

To integrate the information supplied by conventional techniques, retromode imaging has been introduced in several retinal diseases, such as full-thickness macular holes, epi-retinal membranes, and macular edema due to polypoid choroidal vasculopathy, myopic retinoschisis, and subretinal drusenoid deposits, to provide topographic information on RPE alterations [16,17,18,19,20,35,36,37]. 

In a study by Shin and Lee, retromode imaging demonstrated efficacy in detecting fine layers of SRF and RPE alterations in CSC eyes, while these features could not be detected by either FA or ICGA [20]. Furthermore, the potential role of RM imaging in this disease was investigated by Giansanti and colleagues, who analyzed AF, RM, and SD–OCT scans in CSC eyes and reported the observed pathological features [38]. Consistent with previous studies, RM imaging emerged as an easier way to detect even slight changes in RPE, deviating from the plane of the RPE/Bruch membrane and identifying alterations that would not exhibit changes in the autofluorescence signal or were too small to be observed by standard retinography and were easily missed with OCT [16,20,38]. Additionally, SLO RM imaging could identify RPE changes even in the presence of SRF, overcoming the issue of retroillumination [38]. 

In our study, we performed a quantitative analysis of CSC eyes, comparing a traditional technique, AF, with a novel RM technique. Our findings did not reveal any significant difference in the extension of CSC features when analyzed by these two procedures. Moreover, we did not observe any significant difference between RM and AF in the detection of RPE atrophy (81.3% vs. 75%; *p* = 0.683). However, RM exhibited a high TPR (91.7%) and NPV (84.6%).

We hypothesized that the pseudo-3D effect likely played a crucial role in this technique, easing the detection of RPE alterations and providing a good topographic representation of RPE changes in a single acquisition. As reported, AF and RM provided a total comprehensive fundus image and were extremely useful in detecting RPE alterations compared to SD-OCT, which is more detailed but less helpful in the topographic analysis [20]. Of note, RM imaging provides more exhaustive anatomic images that favor the observation of RPE changes, while these data are missed by the functional analysis performed with AF [20]. Nonetheless, we still need to improve the sensitivity of SLO–RM imaging, as it has a roughly 50% ability to discriminate true positives.

Of note, AF is able to show not only morphological changes in an already damaged area but also functional alterations of photoreceptors under metabolic stress, particularly when characterized by the accumulation of lipofuscin [38,39,40]. It might be possible that during the performance of our exams, we observed not only SRF but also other metabolic RPE changes.

Our analysis showed that both RM and AF were more reliable in detecting the area of CSC extension than enface. This outcome was likely due to the ability of enface to identify only already damaged areas and not those that were suffering. Additionally, considering that enface analyses a smaller retinal area than AF and RM, it is understandable that some information was missed with this technique.

According to the heatmap (Figure 6), there was an inverse correlation between age and visual acuity (rho −0.63, 95% CI −0.83–−0.36; *p* < 0.001). Additionally, age is directly correlated with the three diagnostic techniques, particularly with AF and enface. This finding may support two possible hypotheses. First, aging phenomena themselves can exacerbate an underlying pathological condition. Second, considering the mean age of patients at CSC diagnosis, its manifestations may be more chronic and severe in elderly individuals, who have likely been suffering from this entity for several years. According to our findings, we would expect greater damage in elderly patients due to both a likely higher impairment of the RPE and the potential chronicity of epithelial damage over time [41,42]. Enface images were inversely correlated with visual acuity (rho −0.43, 95% CI −0.68–−0.11; *p* = 0.014), while no significant correlation emerged between visual acuity and either AF or RM. This latter finding could be clinically explained by the fact that enface imaging detected only permanently damaged areas, while the other techniques could also show the changes that could recover [38,39,40,43]. 

Our study had some limitations. First, the small sample size did not allow us to provide generalizable and robust findings. A further restriction was the retrospective study design, as well as the lack of correlation between the performed exams and AF and ICG hyperfluorescent areas. Another limitation was that we could not utilize the enface scans of all patients since the quality index of some scans was not high enough to meet the inclusion criteria. However, our single-center experience suggests that SLO–RM is an accurate diagnostic method compared to traditional autofluorescence, preserving the safety of a noninvasive technique. Moreover, due to the use of a long-wavelength laser, it could be performed even without mydriasis or in patients with significant cataracts. Thus, we believe that the introduction of a new noninvasive imaging method could provide important details regarding new diagnostic approaches.

## 5. Conclusions

AF is still considered a key noninvasive imaging method in CSC. However, RM, with its pseudo-3D effect and enface imaging, could support the evaluation of CSC eyes, easily showing RPE alterations. A multimodal imaging approach is a cornerstone for a correct and complete diagnosis, integrating the values and flaws of different techniques.

## Figures and Tables

**Figure 1 life-13-01407-f001:**
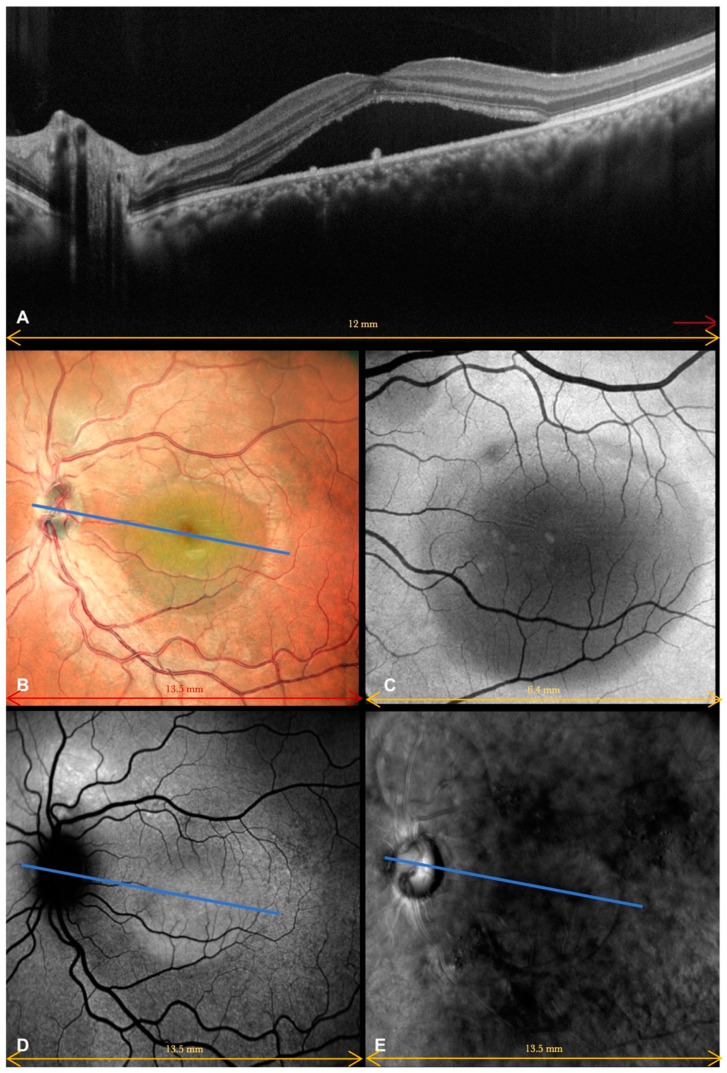
A 30-year-old female patient (best-corrected visual acuity in the left eye: 0.6 LogMar) who suffered from central serous chorioretinopathy assessed by multimodal imaging. (**A**) Spectral-domain optical coherence tomography showed an irregular retinal profile due to the presence of subretinal fluid in the macular region, associated with few retinal pigment epithelium changes (arrows). (**B**) Color fundus retinography clearly showed central serous neuroretinal detachment. (**C**) Central serous detachment was easily observed on enface images as a hyporeflective area in the macular region. The exam could only partially identify a second area of serous detachment above and nasally to the central macular one. (**D**) Autofluorescence showed central serous neuroretinal detachment, which appeared isoautofluorescent, and a hyperautofluorescent area above the optic disc. (**E**) Due to the pseudo-3D effect, retromode imaging clearly detected central serous neuroretinal detachment and a well-outlined secondary serous detachment above the optic disc. RPE mottling was also clearly evident. The blue line in (**B**,**D**,**E**) shows the level where the OCT scan was performed.

**Figure 2 life-13-01407-f002:**
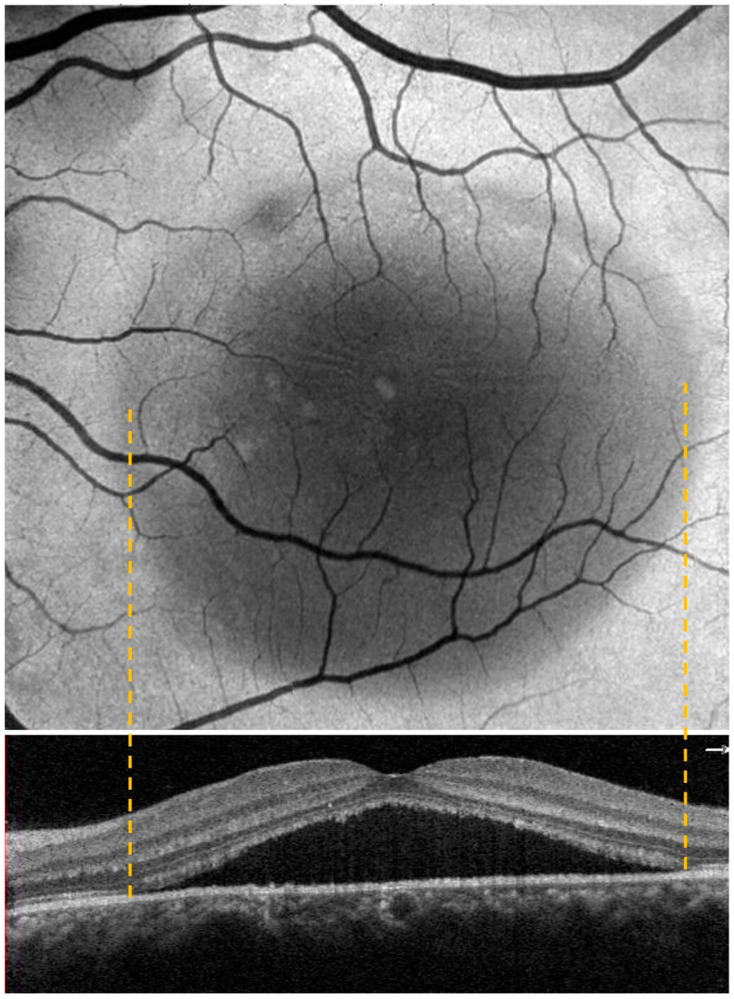
The figure shows the correspondence between the enface and OCT scan at the CSC edge.

**Figure 3 life-13-01407-f003:**
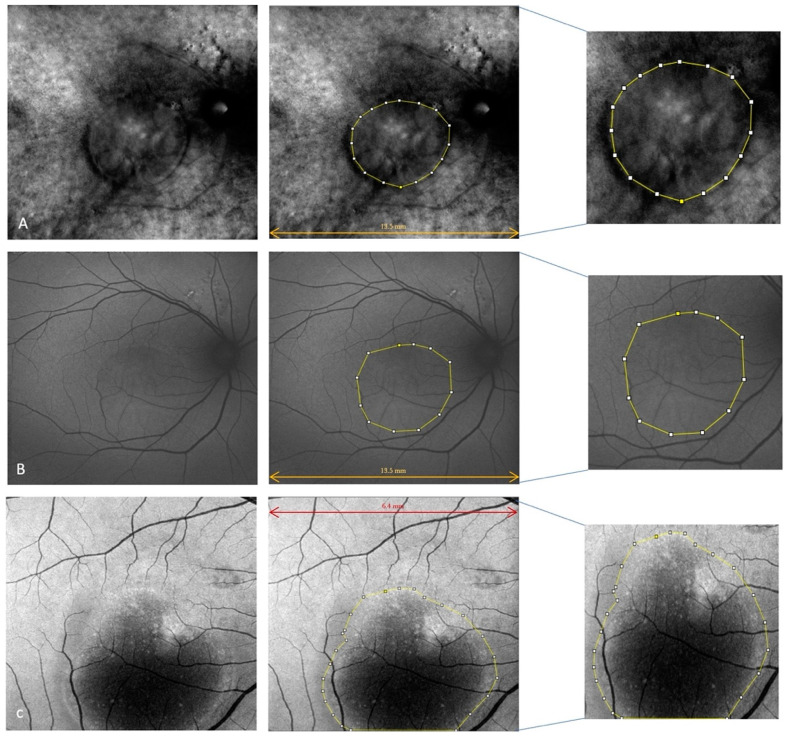
Retromode imaging (**A**), autofluorescence (**B**) and enface (**C**) of the same patient shows serous neuroretinal detachment in the macular region; this was outlined using the provided caliper tool on ImageJ software (on the right).

**Figure 4 life-13-01407-f004:**
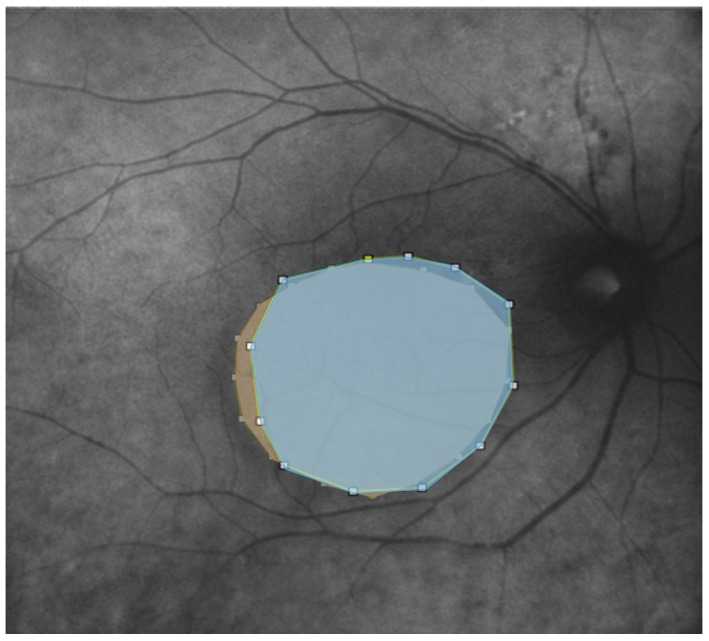
Autofluorescence and Retromode overlaid on top of each other with different colorization (light blue for autofluorescence; yellow for retromode) and transparency to show the differences after using the caliper tool in ImageJ. The pseudo 3D effect of Retromode imaging clearly highlights the outline of serous neuroepithelium detachment in comparison with autofluorescence.

**Figure 5 life-13-01407-f005:**
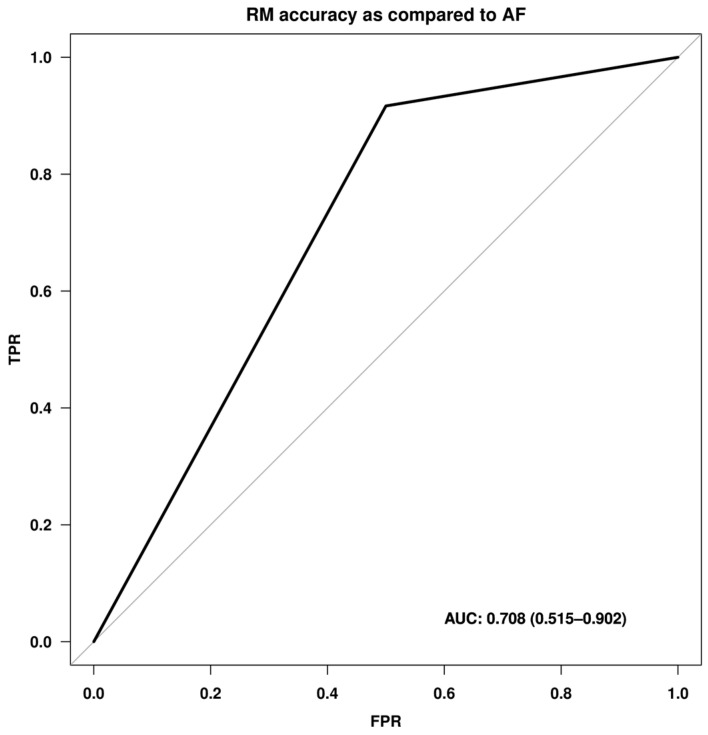
ROC curve comparing the retromode (RM) imaging technique with the standard of care, autofluorescence (AF), in the detection of retinal pigment epithelium (RPE) atrophy. AF was used as the reference line, i.e., RPE atrophy diagnosis by AF was used to assess the accuracy of the diagnosis with the other method, i.e., RM imaging. The displayed values returned a discrete accuracy equal to 70.8% (AUC 0.708). The ordinate axis shows the sensitivity, i.e., the true-positive rate, and the abscissa shows 1—specificity, i.e., the false-positive rate. Abbreviations: FPR: false-positive rate; TPR: true positive rate; AUC: area under the ROC curve.

**Figure 6 life-13-01407-f006:**
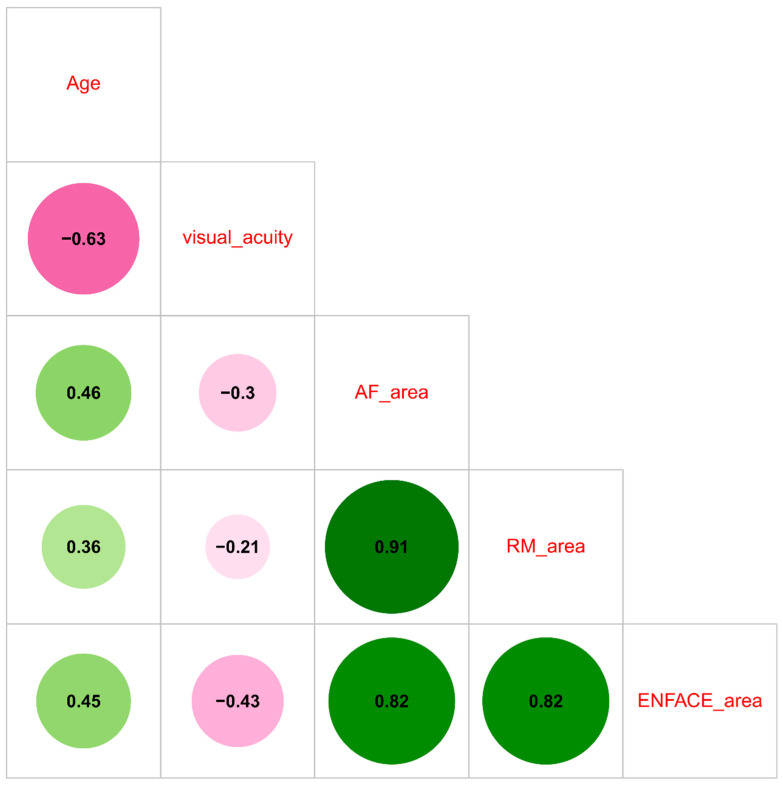
Heatmap showing the correlations among the involved variables (AF: autofluorescence; RM: retromode imaging). Values in green are direct correlations, and values in purple are inverse correlations. The color tones and the size of the circle indicate the degree of correlation.

**Table 1 life-13-01407-t001:** Characteristics of the study sample (N = 32) *.

	N = 32
**Demographics**	
Age (yrs.)	52.7 (13.3)
**Eye**	
Left	21 (65.6)
Right	11 (34.4)
**Visual acuity (LogMar)**	0.69 (0.50–0.90)
**Techniques (mm^2^)**	
Autofluorescence (AF)	19.5 (6.1–29.3)
Retromode imaging (RM)	12.3 (8.1–30.8)
Enface	9.3 (4.8–18.6)
**Diagnosis**	
RPE detachment	6 (18.8)
RPE atrophy by RM	26 (81.3)
RPE atrophy by AF	24 (75)
SRF	30 (93.7)

Abbreviations: RPE: retinal pigment epithelium; SRF: subretinal fluid. * Descriptive statistics are reported as absolute and relative percentage frequency for qualitative data and as the mean and standard deviation (SD) or median and interquartile range (IQR) for quantitative data.

## Data Availability

The data presented in this study are available on request from the corresponding author. The data (original imaging) are not publicly available due to privacy issues.

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
