# Peer review of "Central Serous Chorioretinopathy by Autofluorescence, Enface and SLO–Retromode Imaging"

_life, 2023, doi:10.3390/life13061407_

Round 1
Reviewer 1 Report (Previous Reviewer 1)
The manuscript is acceptable in its current form.
Author Response
Thank you for your revision.
Reviewer 2 Report (Previous Reviewer 2)

The level of English is ok but requires someone to go through and make it sound more precise and scientific. Also the paragraphs including only one sentence should be removed.
Author Response
See the uploaded file

Round 2
Reviewer 2 Report (Previous Reviewer 2)
1. Figure 4 now shows clearly the difference between the AF and RM image segmentations. The authors could elaborate a bit in the caption (or in the text) what is the most likely cause of the difference.
2. Figure 2: I would suggest cropping out the slider bar between the en face image and the B-scan.
3. Also in the figures, it would be useful to have scale bars to get a sense of the scale and size.
Author Response
1. Figure 4 now shows clearly the difference between the AF and RM image segmentations. The authors could elaborate a bit in the caption (or in the text) what is the most likely cause of the difference. We added a sentence of explanation in the caption of figure 4. We added the following sentence on page 8 lines 189-190: The pseudo 3D effect of Retromode imaging clearly highlights the outline of serous neuroepithelium detachment in comparison with autofluorescence. 2. Figure 2: I would suggest cropping out the slider bar between the en face image and the B-scan. Thank you for your suggestion. We changed it. We changed figure 2. 3. Also in the figures, it would be useful to have scale bars to get a sense of the scale and size.We added a line with the indication of image size, since the exported scans are not provided of a scale bar.
We changed figures 1 and 3.
This manuscript is a resubmission of an earlier submission. The following is a list of the peer review reports and author responses from that submission.
Round 1
Reviewer 1 Report
Retromode imaging has been used in previous studies. The introduction and discussion are too elaborate and not relevant to the topic. It has to be precise and relevant to the aim of the study. There are extensive language errors which need correction. The en face images are available in only 18 out of 32 eyes. This needs to be mentioned as a limitation of the study. Therefore comparison between groups so grossly different in size is not a robust comparison. I feel that RPE dystrophy needs to be replaced by RPE atrophy in the whole text and tables. The units of measurements have not been mentioned in the tables. The article needs major revision and shortening of length to make it relevant and interesting. Major revisions are required in the language as well.
